# Compressing and Recovering Short-Range MEMS-Based LiDAR Point Clouds Based on Adaptive Clustered Compressive Sensing and Application to 3D Rock Fragment Surface Point Clouds

**DOI:** 10.3390/s24175695

**Published:** 2024-09-01

**Authors:** Lin Li, Huajun Wang, Sen Wang

**Affiliations:** 1Key Laboratory of Earth Exploration and Infomation Techniques of Ministry of Education, Chengdu University of Technology, Chengdu 610059, China; lilin@stu.cdut.edu.cn; 2College of Mathematics and Physics, Chengdu University of Technology, Chengdu 610059, China; 3Department of Information and Communication Engineering, Tongji University, Shanghai 200092, China

**Keywords:** vibrational signals, MEMS-based LiDAR, rock fragment surface 3D point cloud, clustered compressive sensing, local clustering

## Abstract

Short-range MEMS-based (Micro Electronical Mechanical System) LiDAR provides precise point cloud datasets for rock fragment surfaces. However, there is more vibrational noise in MEMS-based LiDAR signals, which cannot guarantee that the reconstructed point cloud data are not distorted with a high compression ratio. Many studies have illustrated that wavelet-based clustered compressive sensing can improve reconstruction precision. The *k*-means clustering algorithm can be conveniently employed to obtain clusters; however, estimating a meaningful *k* value (i.e., the number of clusters) is challenging. An excessive quantity of clusters is not necessary for dense point clouds, as this leads to elevated consumption of memory and CPU resources. For sparser point clouds, fewer clusters lead to more distortions, while excessive clusters lead to more voids in reconstructed point clouds. This study proposes a local clustering method to determine a number of clusters closer to the actual number based on GMM (Gaussian Mixture Model) observation distances and density peaks. Experimental results illustrate that the estimated number of clusters is closer to the actual number in four datasets from the KEEL public repository. In point cloud compression and recovery experiments, our proposed approach compresses and recovers the Bunny and Armadillo datasets in the Stanford 3D repository; the experimental results illustrate that our proposed approach improves reconstructed point clouds’ geometry and curvature similarity. Furthermore, the geometric similarity increases to 0.9 above in our complete rock fragment surface datasets after selecting a better wavelet basis for each dimension of MEMS-based LiDAR signals. In both experiments, the sparsity of signals was 0.8 and the sampling ratio was 0.4. Finally, a rock outcrop point cloud data experiment is utilized to verify that the proposed approach is applicable for large-scale research objects. All of our experiments illustrate that the proposed adaptive clustered compressive sensing approach can better reconstruct MEMS-based LiDAR point clouds with a lower sampling ratio.

## 1. Introduction

LiDAR technology has revolutionized data collection by providing highly accurate 3D coordinates with small sampling intervals. This capability allows the complex surface features of objects on the ground to be fully captured, overcoming the limitations of 2D images [1]. As a result, short-range LiDAR point cloud data are now widely used to analyze the surface features of rock fragments [2]. Unfortunately, the cost of procuring high-resolution LiDAR hardware impedes the popularity of this application. Recently, advanced Micro-Electro-Mechanical Systems (MEMS) technologies have significantly improved the manufacturing process of various sensors. These ultrasensitive MEMS-based biosensors have been applied to control pandemics at the very initial stage [3]. In the research field of the dynamic pull-in instability of a microstructure, the fractal MEMS systems are suggested to be closer to the actual state as a practical application in the air with impurities or humidity [4]. MEMS-based LiDAR employs a micro-galvanometer with only one emitter to control the light beam [5], making it more reliable due to its simple optical path structure and fewer moving parts. Therefore, short-range MEMS-based LiDAR has significantly reduced the cost of LiDAR hardware from tens of thousands of US dollars to only a few hundred. In addition, MEMS can provide dense point cloud sampling. Comparing expensive mechanical LiDAR for sampling significant geo-objects such as rock outcrops [6], short-range MEMS-based LiDAR is suitable for sampling dense point clouds of rock fragments. However, the large size of the resulting point cloud data leads to high storage and transfer costs. To address this issue, compression based on reduced data size is an available approach for reducing costs.

There are multiple methods for compressing point cloud data, which include traditional techniques such as Huffman-based lossless coding [7], wavelet-based coding [8], and independent component analysis coding [9]. These methods require complete sampling. Alternatively, compressive sensing is more suitable for compressing and recovering high-dimensional data, as it can obtain essential information through random samples.

Mechanical LiDAR has an advantage in producing signals with minimal vibrational noise. Xu R. et al., observed that each dimensional signal is very sparse under the Haar wavelet when sampling point clouds of a broad-leaved tree [10]. Therefore, they proposed a wavelet-based efficient compressive sensing approach to compress each dimension signal of 3D point clouds from a definition Velodyne HDL-32E LiDAR sensor (Velodyne Lidar, Inc., San Jose, CA, USA). MEMS-based LiDAR has more vibrational noise, violating the signal’s sparsity and making it unavailable when using the compressive sensing approach. There are several methods to address this issue. One approach is to utilize machine learning to identify spatial basis functions. For instance, a deep convolutional neural network model can be trained as a basis function. As a result, the stored data contain convolutional kernels of the model and compressed data [11]. Alternatively, a data-driven overcomplete dictionary learning method [12] can be employed to estimate a spatial basis function. This involves learning an overcomplete dictionary and sparse signals through the interior point method. The learned overcomplete dictionary is then utilized for compressive sensing. The final stored data consist of an overcomplete dictionary and compressed data. While the former requires lengthy training and additional storage space for a fixed LiDAR camera, it is well-suited for use in various applications on fixed devices (i.e., fixed-varied). The latter consumes more storage space to maintain the overcomplete dictionary.

Another approach is clustered compressive sensing, which clusters data and applies compressive sensing in each cluster. This method is particularly effective for compressing and recovering vibration signals in wireless network transmission [13]. It considers that even though the sparsity of vibrational signals cannot satisfy the RIP (Restricted Isometric Property) [14] of compressive sensing, sparsity can be achieved within each subcluster. This approach uses the *k*-means algorithm [15] to estimate all clusters. However, the determination of the meaningful *k* value is challenging. An EM (Expectation-Maximization)-based GMM (Gaussian Mixed Model) can be employed to estimate the approximate number of clusters, though this approach may struggle with outliers [16]. Outliers lead to more redundant clusters, meaning that the reconstructed signal has more voids. To address this issue, the DBSCAN clustering algorithm [17] is recommended for estimating the number of clusters closer to the actual [18]. However, the algorithm’s complexity is about $O(n^{2})$, which is a fatal challenge when dealing with the large-scale datasets commonly encountered in real-world scenarios [19]. Many index-based clustering algorithms, such as Silhouette [20] (Sil), Calinski–Harabasz [21] (CH), and Davies–Bouldin [22] (DB), are influenced by the geometric shape of the data. This which can lead to underestimating the number of clusters [23], resulting in distortions in the reconstructed data. Both contributions demonstrate that the sparsity of a subcluster can satisfy the required RIP of compressive sensing. The latter also points out that sparse time–frequency distributions can be applied for each subcluster, as they have been solidified into most systems, such as the wavelet transformation bases. Comparing neural net models and overcomplete dictionaries, a general wavelet transformation basis can save storage space, making it suitable for various devices and applications (i.e., varied–varied), including MEMS-based LiDAR.

Theoretically, the two-scale fractal theory [24] states that two scales can estimate the number of clusters; one is the scale mapping to high-dimensional data self, while the other is the observation scale mapping to an observation point with the same dimensions. Inspired by this, our innovation proposes a local clustering approach based on an observation distance GMM and density peaks to obtain the number of clusters more accurately. Then, clustered compressive sensing is utilized to compress and recover vibrational signals by random sampling as well as for the compression and recovery of MEMS-based LiDAR point clouds. Finally, we improve the recovery precision when the ADMM-BP (Basis Pursuit Algorithm Based on the Alternating Direction Method of Multipliers) algorithm reconstructs the signal with vibrational noise. In conclusion, our contributions are as follows:We find that the clustered compressive sensing approach can improve the performance of compression and reconstruction in point cloud data from a short-range MEMS-based LiDAR camera.We propose a local clustering analysis approach to obtain a closer actual number of clusters automatically based on the observation distances GMM and density peaks.We design a CCS framework to implement MEMS-based LiDAR point cloud compression and recovery.

## 2. Materials and Methods

The proposed method is presented in Figure 1, which outlines the overall process. The study encompasses seven steps: (1) preprocessing point cloud data by simplification, removing outliers, and registering point clouds captured from different angles; (2) local clustering analysis to determine the number of clusters based on the observation distances GMM and density peaks; (3) dividing the complete point cloud data into k clusters using the *k*-means algorithm; (4) reshaping each dimension vector of subcluster data into an N×M matrix; (5) converting each reshaped matrix into a sparse one using a specific wavelet basis; (6) generating the compressed matrix through partial Fourier matrix downsampling; and (7) reconstructing the sparse signal using the ADMM-BP algorithm.

### 2.1. Study Devices, Data Collection, and Data Preprocessing

This study utilized an Intel L515 LiDAR camera (Intel, Inc., Chandler, AZ, USA). to collect point cloud data from rock fragment surfaces. This camera collects the original data based on video streaming technology, with a deep measurement range of 0.6∼9 m and a maximum frame frequency of 30 f/s. Each frame comprises deep distances, a high-definition RGB image, a deep image, and an infra-image. Figure 2 clearly illustrates the three types of images; Figure 2a shows a high-definition RGB image, Figure 2b shows a depth image, and Figure 2c shows an infra image.

Partial point cloud data relate to a single shooting angle. Figure 3 shows partial point clouds captured from various shooting angles.

Benefiting from the high resolution of the high-definition RGB image, this study employed the Scale Invariant Feature Transform (SIFT) algorithm [25] to identify corresponding points across images accurately. Figure 4 shows corresponding points in two different images.

To merge partial point clouds into a complete point cloud, this study utilized the Four-Point Congruent Set (4PCS) algorithm [26] to implement coarse registration and the Iterative Closest Point (ICP) algorithm [27] for fine registration. Figure 5 shows partial point cloud states before and after registration; Figure 5a shows the original data, Figure 5b shows the coarse registered data, and Figure 5c shows the fine registered data.

As examples, in this study three scales of rock fragment samples were used for analysis and experimentation: 32cm×24cm×25cm, 25cm×21cm×20cm, and 10cm×8cm×10cm. Figure 6 shows their complete point clouds.

### 2.2. Motivation for CCS

Compressed sensing theory describes an approach to finding an isometric mapping relation between sensing data and sparse data, as expressed in Equation (Equation 1):(1)y=θ·s
where *y* stands for the sensing data and *s* stands for the sparse data. A convex optimization algorithm can solve the sparse data *s* if the θ is almost isometric. Compressed sensing can improve the reconstruction precision using the same sampling ratio in a cluster. We assume that the whole sampling space is uniform and let *E* be the space energy; *k* invariant subspaces compose the whole data space, and each subspace energy is $E_{k}$, $E_{k}\leq E$. According to the Bayesian theorem (see Equation (Equation 2))
(2)$$P_{X,Y}\left(x,y\right)=P_{X|Y}\left(x|y\right)\cdot P_{Y}\left(y\right),$$
letting $A_{i}$ be the event of acquiring the *i*th isometric vector, if there are *p* isometric vectors, then the energy of each isometric subspace is E/p. Therefore, Equation (Equation 3) shows the probability of finding the matrix θ consisting of p isometric observation vectors in the whole data space:(3)$$P_{A_{1},\cdots ,A_{p}}\left(a_{1},\cdots ,a_{p}\right)=\prod_{i=1}^{n}P_{A_{i}|A_{1},\cdots ,A_{i-1}}.$$
According to the probability theorem, $P_{A_{i}|A_{1},\cdots ,A_{i-1}}=1/\left(E-\left(i-1\right)\cdot E/p\right)$; thus, we can substitute it into Equation (Equation 3) and obtain the formula in Equation (Equation 4): (4)$$P_{A_{1},\cdots ,A_{p}}\left(a_{1},\cdots ,a_{p}\right)=\prod_{i=1}^{n}1/\left(E-\left(i-1\right)\cdot E/p\right).$$

Because $E_{k}\leq E$, the probability of finding *p* isometric vectors in the whole space is less than in each subspace.

### 2.3. Adaptive *k* Parameter in the *k*-Means Algorithm

Utilizing the *k*-means algorithm to obtain clusters is common; however, determining the optimal *k* parameter can be challenging. Our proposed solution involves a local clustering approach that allows for the automatic determination of the *k* parameter based on an observation distances GMM and density peaks.

#### 2.3.1. Local Clustering Based on Observation Distances GMM and Density Peaks

Local clustering algorithms take a small set from the whole data as “seed vertices” and return a good cluster according to the proportion of this small set in the entire data. Other vertices do not confound algorithms for local clustering, as they do not need to be assigned to a cluster.

A. Motivation for using observation distances.

Traditional GMM clustering has two issues, namely, high time costs and sensitivity to outliers. To solve these, we consider the entire dataset as a terrain, with the clusters as hills. The distance between a hill’s top and foot is used to recognize it as a hill. Figure 7 displays the hypothesis.

We randomly take an observation point $o\in R^{m}$ and estimate the distance $d_{i}$ between *o* and the other point $x_{i}\in R^{m}$ in the entire data space $X\in R^{m}$, as expressed in Equation (Equation 5):(5)$$d_{i}=\sqrt{{∥o-x_{i}∥}_{2}^{2}}.$$

All distance scalars compose a distance vector $D=\left[d_{1},\cdots ,d_{n}\right]$, and the dense region of distances is considered to be a cluster. The high-dimensional data are converted to a one-dimensional vector, avoiding the calculation of multiple covariances and influence by outliers.

B. Estimating the optimal GMM based on observation distances.

Our approach iteratively estimates (N−1) GMMs using the EM algorithm based on observation distances. It selects the GMM with the minimum AIC (Akaike information criterion) value [28] as the optimal solution, treating the Gaussian component number of the optimal GMM as the candidate cluster number. However, the optimal GMM is sensitive to the observer’s location. Figure 8 shows different GMMs for different observers. Observer A finds a one-component GMM in the middle of two datasets, while Observer B finds a two-component GMM.

Although components of the optimal GMM cannot directly determine clusters, they provide a better Probability Density Function (PDF) for reducing low-probability data. Our approach involves selecting the top 30% of distance probability data, or 5% for larger point cloud sizes, and utilizing density peaks to extract potential clusters in the reduced dataset $X_{c}$.

C. Finding potential cluster centers using density peaks.

Rodriguez and Laio proposed a clustering approach based on density peaks and one-step estimation [29]. This approach first estimates the density of each point by counting points in a circle with the data point as the center and a cutoff distance $d_{c}$ as the radius, then finds the highest density points as the center of clusters. In our study, we employ a Gaussian density function instead of simply counting points. Equation (Equation 6) calculates the local density $\theta_{j}$ of the *j*th point $x_{j}$ ($x_{j}\in X_{c}$):(6)$$\theta_{j}=\sum_{i=1}^{n}e^{-{\left({∥x_{j}-x_{i}∥}_{2}/d_{c}\right)}^{2}}.$$

Next, the proposed method involves estimating the density distance of each point through a two-step process: first sorting the local densities of all points in ascending order ($\theta_{1}\leq \theta_{2}\leq \ldots \leq \theta_{n}$), then calculating the density distance, which is the minimum distance between the current point and the other point corresponding to the higher density, as expressed in Equation (Equation 7):(7)$$d_{\theta_{j}}=\min_{i=j+1,\ldots ,n}{∥x_{j}-x_{i}∥}_{2}.$$

For the point with the highest density, the density distance is conventionally calculated using Equation (Equation 8):(8)$$d_{\theta_{n}}=\max_{i=1,\ldots ,n-1}{∥x_{n}-x_{i}∥}_{2}.$$

The densities and density distances of all points compose a decision graph. Letting $d_{c}$ be 5, Figure 9 shows this graph for the third component of the optimal GMM in the KEEL [30] Wine-White dataset.

Candidate cluster centers are points with higher density and density distance. We can roughly set the density distance threshold to 30%, as shown in the red line of Figure 9. Those points in the top right-hand quadrant are candidate cluster centers. In this way, candidate cluster centers accumulate in all components of the optimal GMM.

D. Determining cluster centers using inflection points of dissimilarity.

Certain redundant candidate cluster centers, such as identical and closer points, must be removed; given *m* candidate cluster centers, the dissimilarity is the Euclidean distance between any two candidate cluster centers. In contrast to removing points with a threshold, we remove redundant candidate cluster centers with inflection points [31] in the dissimilarities sequence. The inflection points method is more interpretable. It can solve the inflection points with a second-order difference of zero, then merge all points with dissimilarity smaller than the inflection point. Finally, the merged number of candidate cluster centers is the estimated number of clusters, i.e., the *k* parameter.

#### 2.3.2. Algorithm and Complexity Analysis

Algorithm 1 shows the local clustering algorithm based on the observation distances GMM and density peaks.

Algorithm 1 shows that all calculations are one-step. Given a dataset *X* with *N* objects, the algorithm complexity is estimated through the following steps:Step1: Calculate the distance distribution. The computational complexity is O(dN).Step2: Solve the best GMM. Given *K* Gaussian components of a GMM, the expectation value algorithm complexity is O(2NK). Letting the average iteration of the maximization algorithm to α, the EM algorithm complexity is O(2αNK); letting the maximum number of the Gaussian components be $K_{max}$, the general complexity is $O(2\alpha NK\left(K_{max}-1\right))$.Step3: Use density peaks to determine candidate cluster centers. Given $N_{c}$ high-probability data, the density estimation complexity is $O\left(N_{c}\right)=\left(N_{c}\times \left(N_{c}-1\right)\right)/2$.Step4: Remove redundant candidate cluster centers. Letting *S* be the number of inflection points in the dissimilarity set and *M* the average number of dissimilarities less than the inflection points, the complexity is O(S)=M×S.

Thus, the total computational complexity is $O\left(dN\right)+O\left(2\alpha NK\left(K_{max}-1\right)\right)+O\left(N_{c}\right)+O\left(S\right)$.

### 2.4. Point Cloud Compression and Recovery Based On CCS

The k-means algorithm conveniently obtains all clusters according to our estimated k parameter. The spatial coordinates of each cluster are split into three one-dimensional vectors, which are labeled as [X], [Y], and [Z]. After reshaping each vector to a matrix consisting of $X_{M}$, $Y_{M}$, and $Z_{M}\in R^{256\times n}$, we apply the Discrete Wavelet Transform (DWT) to these matrices to make them sparser. Subsequently, we utilize the partial Fourier matrix to implement downsampling. Finally, the ADMM-BP algorithm is employed to reconstruct the sparse matrices and the inverse DWT is used to recover the original coordinates.

**Algorithm 1** Local clustering algorithm based on observation distances GMM and density peaks**Input:** Given data set X∈Rm×n. Maximum size of components in a GMM Mmax. Likelihood convergent threshold τ.**Output:** Cluster centers set *C*.
  1:Randomly and uniformly initialize the observation point O∈Rn in *X*. Cluster centers set C=Φ, Φ is the empty set.  2:Calculate distance vector *D* as Equation (Equation 5).  3:Estimate the best GMM with the EM algorithm:  4:**for** M := 2 to Mmax **do**  5:    The EM algorithm solves the parameters of each component in the *M*th GMM.  6:    Calculate the AIC value for the *M*th GMM.  7:**end for**  8:Choose the best GMM with the minimum AIC value.  9:Given *K* components of the best GMM. Calculate candidate cluster centers with the density peaks.10:**for** i := 1 to K **do**11:    Extract the top 10% of data Xc based on the probability of the *i*th component.12:    Extract high-density points as candidate cluster centers and append to set *C* as Equations (Equation 6)–(Equation 8).13:**end for**14:Given *M* candidate cluster centers set C{c1,…,cM}, and Remove redundant cluster centers:15:**repeat**16:    Calculate the dissimilarity set Dc{d1,…,dM×(M−1)/2} and Sort the set Dc in an ascending order.17:    **if** There are inflection points in the set Dc **then**18:        Merge the cluster centers with dissimilarity values smaller than inflection points.19:        Update the number *M* of cluster centers and clustering centers *C*.20:    **end if**21:**until** No the inflection points in the ordered dissimilarity set {d1,…,dM×(M−1)/2}


#### 2.4.1. Sparse Transformation

Figure 10 shows the distributions of the three dimensions in a point cloud of rock fragment surfaces.

Figure 10a shows that more vibrational noise along the X dimension cannot make the wavelet-transformed signal more sparse [14]. After conducting comparative experiments with other wavelets, it was found that the coif1, db2, and bior1.1 wavelets yield more minor reconstruction errors for the X, Y, and Z dimensions. The waveforms of these three wavelets are represented in Figure 11.

Following the principles of compressive sensing theory, the variables $X_{M}$, $Y_{M}$, and $Z_{M}$ are subjected to sparse transformation as delineated in Equation (Equation 9). For illustrative purposes, $X_{M}$ is examined as a representative example:(9)$$X_{S}=\Psi X_{M},$$
where Ψ denotes the discrete wavelet matrix with eight decomposition levels.

#### 2.4.2. Data Downsampling

Compressed sensing provides an alternative paradigm to sampling theory [32]. Generally, a signal *x* with sparsity k ($x\in R^{N}$) can be reconstructed by the linear measurement matrix $A\in R^{M\times N}\left(M\ll N\right)$. The formula can be written as y=Ax+n, where $n\in R^{M}$ can be assumed as Gaussian noise with variance $\sigma_{n}^{2}$. If matrix *A* satisfies the RIP principle, then the signal *x* can be reconstructed from M=O(Klog(N/K)) samples [33]. In this study, we generate the compressed data $X_{C}$ using the product of a partial Fourier matrix and a sparse matrix. Equation (Equation 10) shows the computing process:(10)$$X_{C}=\Phi X_{S}$$
where Φ denotes the measurement matrix and $X_{C}$ represents the compressed matrix. Roughly speaking, measurements are good if they are incoherent concerning the columns of the sparsifying basis [34], meaning that the rows of the measurement matrix have a small inner product with the columns of the sparsifying basis. The coherence can be utilized to estimate the incoherence, with greater coherence/lower incoherence indicating that fewer rows are required in the measurement matrix and vice versa [35]. In our experiments, we calculate the coherence between different measurement matrices and the sparsifying basis using Equation (Equation 11):(11)$$\mu \left(C,\Psi \right)=\sqrt{n}\underset{j,k}{max}\left|<c_{k},\psi_{j}>\right|$$
where μ(C,Ψ) denotes the coherence between the measurement matrix and the sparsifying basis, $c_{k}$ indicates the *k*-th row in the measurement matrix, and $\psi_{j}$ indicates the *j*-th column in the sparsifying basis.

The comparison of coherence illustrates that the partial Fourier matrix has the highest coherence value. Therefore, we consider the partial Fourier matrix as our measurement matrix, as shown in Figure 12.

#### 2.4.3. Data Reconstruction

Compressive sensing theory includes two approaches for reconstructing the primary signal, namely, convex optimization methods and greedy algorithms. The former reconstructs the primary signal through norm minimization; the ADMM-BP algorithm is one of the most important algorithms of this kind [36]. In the latter approach, the locations of nonzero elements are estimated, then the amounts of these elements are calculated. The Matching Pursuit (MP) and Orthogonal Matching Pursuit (OMP) algorithms are among the most popular and widely used greedy algorithms. Each of the two reconstructing approaches has advantages and disadvantages. Generally, optimization methods guarantee exact signal reconstruction, but have much lower speed compared to greedy approaches. This study uses the ADMM-BP algorithm to reconstruct the primary signal using $l_{1}$ minimization, a solving approach that benefits from the convergency of the Lagrange multiplier method and the decomposition of the dual ascent method.

Based on an equality constraint, the ADMM-BP algorithm solves the $l_{1}$ minimization problem following Equation (Equation 12):(12)$${minimize∥x∥}_{1}\;s.t.Ax=b$$
where $x\in R^{n}$ denotes the sparse data, $A\in R^{m\times n}$ stands for the measurement matrix, and $b\in R^{m}$ represents the measurements. The above variables satisfy m<n. The following steps describe iterative updating in the ADMM-BP algorithm:

Step 1: Update *x* using Equation (Equation 13):(13)$$x_{k+1}:=P\left(z_{k}-y_{k}\right)$$
where *P* denotes a projection function on $x\in R^{n}|Ax=b$, $z_{k}\in R^{n}$ stands for Lagrange multipliers, and $y_{k}\in R^{n}$ stands for the duals of the Lagrange multipliers.

Step 2: Update *z* using Equation (Equation 14):(14)$$z_{k+1}:=1/\rho \left(x_{k+1}+y_{k}\right)$$
where ρ stands for the Lagrange augmentation parameter.

Step 3: Update *y* using Equation (Equation 15):(15)$$y_{k+1}:=y_{k}+x_{k+1}-z_{k+1}.$$
Updating *x* involves a Euclidean minimized norm problem based on a linear constraint, as shown in Equation (Equation 16):(16)$$x_{k+1}:=\left(I-A^{T}{\left(AA^{T}\right)}^{-1}A\right)\left(z_{k}-y_{k}\right)+A^{T}{\left(AA^{T}\right)}^{-1}b.$$

Considering that the measurement and compressed matrices are complex numbers, we can improve the $l_{1}$ minimization process in the traditional ADMM-BP algorithm by replacing the real vector with a complex number matrix. Algorithm 2 shows the enhanced ADMM-BP algorithm for $X_{C}$:

**Algorithm 2** Improved ADMM-BP algorithm**Input:** Compressed matrix XC∈Cm×nXC, Measurement matrix Φ∈Cm×n, Augment Lagrange parameter ρ∈R, and over relax parameter α∈R. m≪n.**Output:** Reconstructed sparse matrix S∈Cn×nXC.  1:Initialize the maximum iteration number MAX_ITER=1000, the Lagrange multipliers matrix Z∈Rn×nXC=ϕ, and the duals of Lagrange multipliers matrix U∈Cn×nXC=ϕ, where ϕ denotes empty set, *C* for complex number set. AAt=ΦΦH, P=I−ΦH(ΦΦH)−1Φ, Q=ΦH(ΦΦH)−1XC,i=1,ABSTOL=1e−4,RELTOL=1e−2,S=ϕ.  2:**while** i≤MAX_ITER **do**  3:    Updating *S*: S=P(Z−U)+Q, Zold=Z  4:    Over relaxing for *S*: S=αS+(1−α)Z  5:    Updating *Z*: Z=1/ρ(S+U)  6:    Updating *U*: U=U+(S−Z)  7:    Skip conditions:  8:    **if** ∥S−Z∥2<(nABSTOL+RELTOL·MAX(∥S∥2,∥−Z∥2)) and ∥−ρ·(Z−Zold)∥2<nABSTOL+RELTOL·∥ρ·U∥2 **then**  9:        break:10:    **end if**11:    i=i+112:**end while**

To compute the $l_{1}$ norm in a matrix instead of a vector, we convert each row vector $s_{i}^{H}$ of *S* to $∥s_{i}^{H}{∥}_{2}$. Observing the iteration process of S+U in updating *Z*, most energies are distributed at a low frequency, and these low-frequency data have a tiny imaginary-number part. In this study, we take the real-number part to update *Z*. The iterations of the ADMM-BP algorithm depend on the precision threshold. Fewer samplings leads to more iterations and lower reconstruction speed.

## 3. Results

To better illustrate the availability of our proposed adaptive CCS approach, our experiments consisted of local clustering experiments, experiments on Stanford public 3D point clouds [37], experiments on point clouds of rock fragment surfaces, and experiments on the point cloud of a rock outcrop. The results of the local clustering experiments on four public datasets from the KEEL repository verify the validity of our proposed approach, while experiments on the Stanford 3D point clouds demonstrate its generality. In addition, we implemented compression and recovery of the rock fragment surface point cloud data based on our datasets, following an analysis of the wavelet types of Intel L515 LiDAR camera signals. Finally, experiments on a rock outcrop point cloud indicate that our method is applicable to a large-scale point cloud collected via mechanical LiDAR. We implemented our algorithms in MATLAB R2022a. The testing platform was an Intel^TM^ Core^®^ I5-11300H CPU @ 3.10GHZ, 16 GB RAM, and Windows 11 operating system.

### 3.1. Local Clustering Experiments

Our team conducted experiments on four KEEL datasets containing more than 1500 samples using our local clustering algorithm. The correct class number was listed to ensure clarity. Compared to the GMM, Silhouette (Sil), Calinski–Harabasz (CH), and 350 Davies–Bouldin (DB) analysis, our approach provides an estimated number of clusters that is closer to the actual number.

A. Experimental settings.

In our algorithm, the observation distance GMM is utilized to estimate the PDF of all data. In this experiment, we take the top 45% of high-likelihood data for data sizes less than 5000 and 30% between 5000 and 10,000. In comparison, only 5% are selected for sizes larger than 10,000. The density and density distance thresholds are set to 30% and the cut-off distance $d_{c}$ is 5. When extracting inflection points, we set the threshold of the inflection point to $10^{-4}$ instead of 0 to address the approximation error in the second-order difference.

B. Experimental details of the Winequality-white dataset.

The Winequality-white dataset contains 4898 samples and seven classes numbered 3 to 9. The observation distances GMM consists of seven Gaussian components; Table 1 shows each component’s proportion, mean, and variance, while Figure 13 shows the PDF of the GMM.

The decision graphs corresponding to each Gaussian component are generated based on the densities and density distances estimation, as shown in Figure 14.

Figure 14 shows eleven candidate cluster centers in seven Gaussian components. We calculate and sort the dissimilarities between these candidate cluster centers, as shown in Figure 15. Finally, we remove redundant candidate cluster centers with the inflection points method and obtain seven cluster centers.

C. Experiments on multiple public datasets.

We verified our proposed method on four public benchmark datasets from the KEEL repository. The proposed method was compared with the GMM. Table 2 shows the experimental result.

After comparing the experimental results, our proposed method appears more accurate in determining the number of clusters.

### 3.2. Experiments on Stanford Public Point Cloud Datasets

Figure 16 shows various dimensional signal distributions of the Bunny and Armadillo datasets from the Stanford 3D Point Cloud Repository [37]. These diagrams represent more vibration characteristics, meaning that the wavelet-based transformed datasets have insufficient sparsity.

A. Experimental parameter settings.

In the clustering experiment, the maximum number of Gaussian components is 100. We take the top 5% of high-likelihood data based on the PDF of the optimal GMM. The thresholds for the densities and density distances are set to 30%, while the cut-off distance ($d_{c}$) is set to 5. The bior1.1 wavelet is applied to the sparse transformation during the compression process. To better understand the effects of different cluster numbers, the sparsity is set to 0.35. The partial Fourier matrix is considered the measurement matrix for implementing random sampling, and the sampling ratio is 0.4. In the reconstruction process, the ADMM-BP algorithm is utilized to reconstruct the data.

B. Experimental indices settings.

In our local clustering analysis, the maximum component number of the GMM is 100. Based on the observation distances, we take the top 5% of high-likelihood data with the GMM. The density and density distance thresholds are set to 30%, with a cutoff distance $d_{c}=5$. In compression and reconstruction, the bior1.1-based DWT is applied to sparsify the original signals; the sparsity is 0.8. The measurement matrix is the partial Fourier matrix. The reconstruction algorithm is the ADMM-BP algorithm, as it is less affected by the RIP. We assume that the benchmark point cloud data contain no distortions. Full-Reference Point Cloud Quality Assessment (FR-PCQA) provides an excellent evaluation of the reconstructed point cloud data [38]. As the 3D point cloud data structure has three dimensions instead of 2D images, we utilize the Point Cloud Structural Similarity Index Measure (PC-SSIM) and Root Mean Square Error (RMSE) as evaluation metrics. PC-SSIM scripts [39] are used for computing the point cloud structural similarity scores and preprocessing stages (i.e., point fusion, voxelization, and attribute estimation). We select the geometry, normal, and curvature attributes as the measurement indices. The similarity score is a pair of values that contain the minimum variance similarity and minimum mean value similarity compared with the reference data. The sparsity reconstruction error measurement based on the DWT refers to the original point cloud data, while the compressed reconstruction error measurement based on the compressive sensing refers to the DWT-based reconstructed point cloud data.

C. Comparative experiment using various compressive sensing approaches.

Table 3 shows the PC-SSIM and RMSE results for the DWT-based reconstructed point cloud data and the original data. Figure 17 shows that the DWT-based reconstructed point cloud data are similar to the original data.

The number of clusters is estimated separately using our proposed method and the traditional GMM. Table 4 shows the PC-SSIM and RMSE results for the non-clustered, traditional GMM clustered, and proposed CCS method, referred to as DWT-based reconstructed point cloud data. In addition, we compare indices-based CCS methods utilizing the Silhouette, Calinski–Harabasz, and Davies–Bouldin indices as clustering analysis metrics. The sampling ratio is set to 0.4, and “∖” denotes the non-clustered compressive sensing method.

Comparing the minimum mean value of the geometrical similarity in Table 4, it is easy to see that the similarity rises with an increasing number of clusters. Meanwhile, it is important to note that, based on comparison of the curvature similarity, an exceedingly high number of clusters is not necessary. The bunny dataset shows lower curvature similarity (−0.0020) when the number of clusters is 99, and the curvature similarity is $-2.2676\times 10^{-4}$ when the number of clusters is 41 with the DB index. This is because excessive clusters lead to more voids in the reconstructed data. Figure 18 and Figure 19 show pictures of the experimental results. Figure 18d shows that excessive clusters cause more empty holes in the bunny’s ear.

### 3.3. Experiments on Rock Fragment Surface Point Clouds

A. Experimental settings.

An in-depth analysis using Haar, db2, coif1, bior1.1, sym2, rbio1.1, and fk4 was conducted to determine the appropriate wavelet. In this experiment, we selected partial point clouds corresponding to shooting angles $0^{\circ }$ and $180^{\circ }$ from each mentioned scale. To ensure the credibility of our experiment, we only varied the wavelet basis while maintaining the measurement matrix and reconstruction algorithm. For convenience, we set the number of clusters to 30. The optimal wavelet was determined by roughly comparing the RMSE. While the RMSE is not suitable for the comparing the errors of different vectors, it is preferred for comparing the errors of one-dimensional scalars.

Figure 20a shows the coif1 wavelet, which has a small RMSE on the X dimension. Figure 20b shows the db2 wavelet, which has a small RMSE on the Y dimension. Finally, Figure 20c shows the bior1.1 wavelet, which has a small RMSE on the Z dimension.

B. Experiment with our datasets.

After conducting experiments on the measurement matrix and sparsifying wavelet basis, we used the complete point clouds in Figure 6 to verify the proposed approach. The sizes for Figure 6a–c are 864,836, 177,490, and 30,880, respectively. All algorithms were performed in a single thread. Table 5 shows the PC-SSIM and RMSE results.

Our innovative CCS approach outperforms the other methods, as shown in Table 5. The Silhouette approach fails to determine a cluster count for the point cloud data in Figure 6a; thus, all “Silhouette” items are marked with the “∖” symbol. The reconstructed similarity of the geometrical attributes exceeds 0.9, while that of the normal attributes surpasses 0.8. The curvature similarity increases exponentially. Our findings are showcased through a comparison of non-clustered compressive sensing, indices-based CCS, and our proposed CCS approach, presented in Figure 21, Figure 22 and Figure 23. Notably, the X-dimension of the reconstructed data incurs signal distortion when using non-clustered, Sil-based clustered, and CH-based clustered compressive sensing due to an influx of vibrational noise.

### 3.4. Clustered Compressive Sensing Experiments on Large-Scale Outcrop Datasets

Rock outcrops are large-scale research objects in petrophysics. A mechanical LiDAR camera was used to collect an enormous amount of point cloud data on a rock outcrop [40]. In the compressive sensing field, the DWT based on the Haar wavelet and OMP algorithm is suitable for compression and recovery of mechanical LiDAR signals [10]. In this section, we employ the obtained rock outcrop point cloud data to verify the validation of our proposed method. Figure 24 shows the outcrop point cloud data, while Figure 25 shows the shape of each dimensional signal.

Following Xu et al., we set the sampling ratio to 0.4; the sparsifying basis was Haar wavelet, the measurement matrix was the partial Fourier matrix, and the reconstruction algorithm was the OMP algorithm. Table 6 shows the PC-SSIM and RMSE results for both compressive sensing approaches.

In comparing the curvature similarities, the reconstructed point cloud data based on our proposed CCS approach are more similar to the DWT-based point cloud data. The cost time is more than the traditional compressive sensing in single-thread computation; however, our proposed method is more effective in parallel computation.

## 4. Discussion

### 4.1. Insights into MEMS-Based LiDAR Point Cloud Compression and Recovery by CCS

Undoubtedly, compressive sensing is an efficient method for real-time compression and recovery of data based on random sampling. The MEMS-based LiDAR camera provides exact point cloud sampling of rock fragment surfaces with lower acquisition costs. However, due to the internal mechanism of MEMS-based LiDAR, the collected signal is a vibrational signal, and the signal needs to be sparser in the DWT-based signal. Therefore, the traditional compressive sensing approach can only partially reconstruct the DWT-based signal. While earlier studies have explored how clustered compressive sensing can reconstruct the vibrational signal more precisely, they have yet to explicitly address the fact that a reasonable clustering analysis is more valid for reconstructing the point cloud of a MEMS-based LiDAR camera.

Using our proposed CCS approach, we found that the reconstructed signal is more similar to the DWT-based signal. In our experiment with the Stanford Armadillo data, comparing the similarity of the reconstructed signal obtained with traditional compressive sensing and our approach shows that the geometrical similarity increased from 0.3279 to 0.6791 and the curvature similarity rose from $-1.9452\times 10^{-4}$ to 0.0098. In addition, we obtained better wavelets for each dimensional signal in our rock fragment surface point cloud experiments; the geometrical similarity increased to 0.92, the curvature similarity rose to 0.0081, and the RMSE decreased.

Although Ma et al. found that CCS can compress and recover the vibrational signal based on single GMM clustering analysis in the wireless network transfer process, our findings suggest that local clustering analysis based on the observation distances GMM and density peaks is better than a single GMM. Instead of using the simple high-dimensional data points, our proposed local clustering analysis is based on the distances between the observer and the data points. Using distances-based GMM solves the dimensionality disaster in high-dimensional data and estimates all local components. Each local component provides a PDF to reduce the original data. We then calculate the densities and density distances based on the reduced data points and estimate the density peaks using a decision graph that consists of the densities and density distances. Each density peak represents a candidate cluster center. Finally, The number of clusters is obtained after using the inflection point method to remove redundant candidate centers. Our local clustering analysis experiments on four UCI benchmark datasets illustrate that the estimated results are closer to the number of clusters. In our experiments with the Stanford Bunny data, a higher number of clusters leads to more empty holes, resulting in more dissimilarities being produced in the curvature attribute.

Regarding algorithm efficiency, our local clustering algorithms consist of a one-step estimation and are more efficient under parallel computation. In our dataset, as shown in Figure 6a, 864,836 3D coordinates map to 864,836 distances. The distances GMM and density peaks are computable with lower memory and CPU cost. The algorithm is also suitable for GPU computation if higher efficiency is required. While the traditional compressive sensing algorithm runtime is faster than ours under single-thread computation, our proposed CCS approach has a faster runtime speed under parallel computation.

### 4.2. Limitations and Future Work

When using k-means clustering, choosing the value of the *k* parameter that represents the number of clusters is often a significant challenge. Erich Schubert et al. explained that the elbow method cannot obtain a relatively correct number of clusters, and suggested certain GMMs that can better estimate the *k* value in the *k*-means algorithm [41]. In general, there are three types of clustering algorithms for calculating the *k* parameter:(1)*Global clustering algorithms* assign every vertex to one of *k* clusters, which may or may not be disjoint. Algorithms for this problem may be unsupervised, such as SpectralClustering [42], or semisupervised, such as the regional force-based methods [43].(2)*Local clustering algorithms* take as input a small set of “seed vertices” and return a good cluster containing the small set. Background vertices do not confound algorithms for local clustering, as they do not need to be assigned to a cluster. It is possible to further subdivide local clustering algorithms into strong and weak local clustering algorithms; strong local algorithms such as Nibbon [44], PPR-GROW [45], and Capacity-Releasing Diffusion [46] are characterized by having runtimes proportional to the size of the obtained clusters, while weak local algorithms such as the GMM are characterized by having runtimes proportional to the data size, and are frequently faster than strong local algorithms when finding large or moderately sized clusters.(3)*Cut improvement algorithms* take a cut or subset as input, which can be thought of as an approximation of a cluster, then refine it to produce a better approximation.

In this study, we chose a weak local clustering algorithm due to the need for fast computational speed when estimating the number of clusters. Although we utilize the density peaks to find hidden cluster centers missed by a single observer, the limited observation ability of a single observer still cannot accurately identify the actual number of clusters. In [47], the authors proposed a clustering algorithm based on a finite mixture of gamma distributions, illustrating that a mixed gamma model can provide a better probability model for clustering data. However, for big data the likelihood mostly does not converge during the EM algorithm process, and approximate solving has higher time costs when the Euler–Maclaurin formula applies to the Digamma function. Using multiple observers can improve the estimated precision; however, this method leads to exponentially increasing time consumption. The ClusterPursuit algorithm [48] is a more efficient algorithm with an advantage in graph analysis concerning the local clustering field. However, the size of the graph used to generate a network with a size of 10,000 × 10,000 is restricted to 10,000 points, which makes the network is difficult to compute. Our proposed observation distances GMM provides a model with the probability needed to reduce the data reasonably. Based on the reduced data, the ClusterPursuit algorithm may find the number of clusters more accurately.

## 5. Conclusions

This study solves the compression and recovery of a rock fragment surface point cloud from a MEMS-based LiDAR camera using adaptive clustered compressive sensing. Unlike conventional mechanical LiDAR, MEMS-based LiDAR signals are vibrational, meaning that the traditional compressive sensing approach cannot reconstruct the sparse signal due to not satisfying the RIP. We consider that the likelihood of acquiring the isometric vector in a subspace is higher than in the entire space under the same sampling rows conditions; as such, we employ the *k*-means clustering algorithm to obtain subspaces. However, the *k* value, representing the number of clusters, is challenging to estimate. Thus, we propose a local clustering algorithm based on observation distances GMM and density peaks to estimate the closer number of clusters. Our experiments on the UCI repository illustrate that the number of clusters estimated using our approach is closer to the actual cluster number than with the single GMM clustering algorithm. In experiments on the Stanford 3D repository, our proposed CCS approach can reconstruct the Bunny and Armadillo point cloud data, in contrast to the traditional compressive sensing approach, and there are more similarities in the curvature attribute compared to the number of clusters estimated by the single GMM clustering algorithm. Finally, in experiments with our obtained large-scale rock outcrop dataset, the proposed approach is more efficient than traditional compressive sensing.

## Figures and Tables

**Figure 1 sensors-24-05695-f001:**
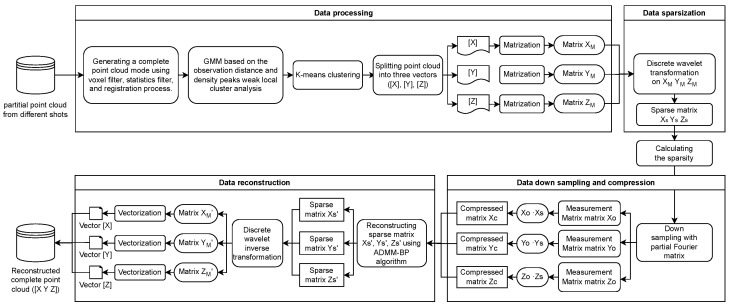
Flow chart of the proposed method.

**Figure 2 sensors-24-05695-f002:**
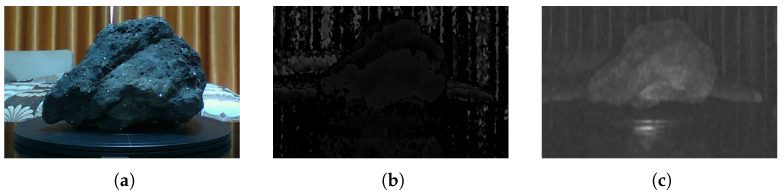
Three types of images taken using the Intel L515 LiDAR camera: (**a**) high-definition image, (**b**) deep image, and (**c**) infra-image.

**Figure 3 sensors-24-05695-f003:**
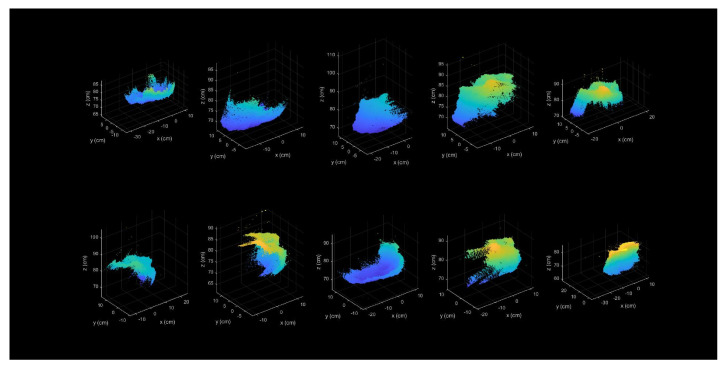
Partial point clouds from various shooting angles.

**Figure 4 sensors-24-05695-f004:**
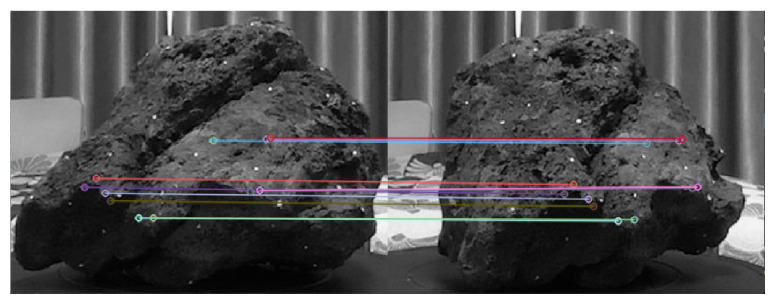
Corresponding points in two different images.

**Figure 5 sensors-24-05695-f005:**
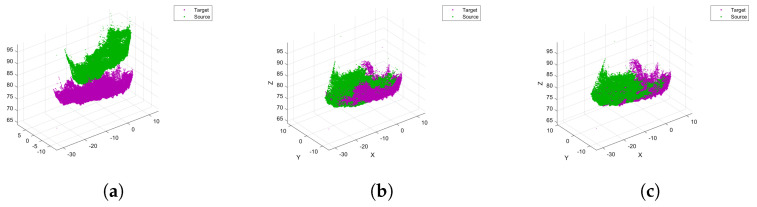
Partial point cloud states before and after registration: (**a**) original partial point clouds, (**b**) coarse registered partial point clouds, and (**c**) fine registered partial point clouds.

**Figure 6 sensors-24-05695-f006:**
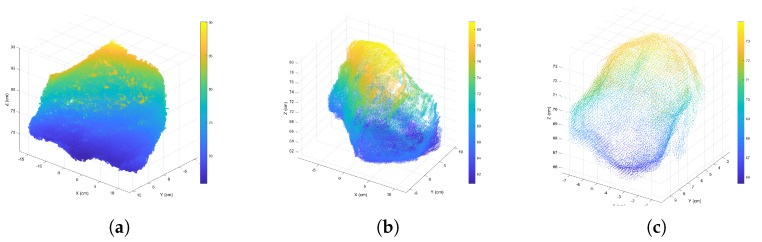
Complete point clouds for rock fragment surface across three scales: (**a**) 32cm×24cm×25cm, (**b**) 25cm×21cm×20cm, and (**c**) 10cm×8cm×10cm.

**Figure 7 sensors-24-05695-f007:**
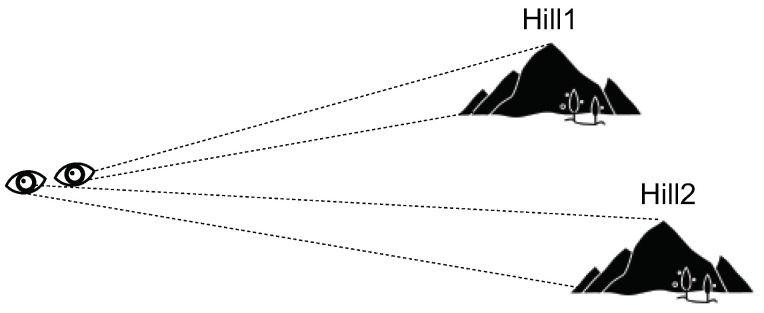
Clustering motivation diagram based on observation distances.

**Figure 8 sensors-24-05695-f008:**
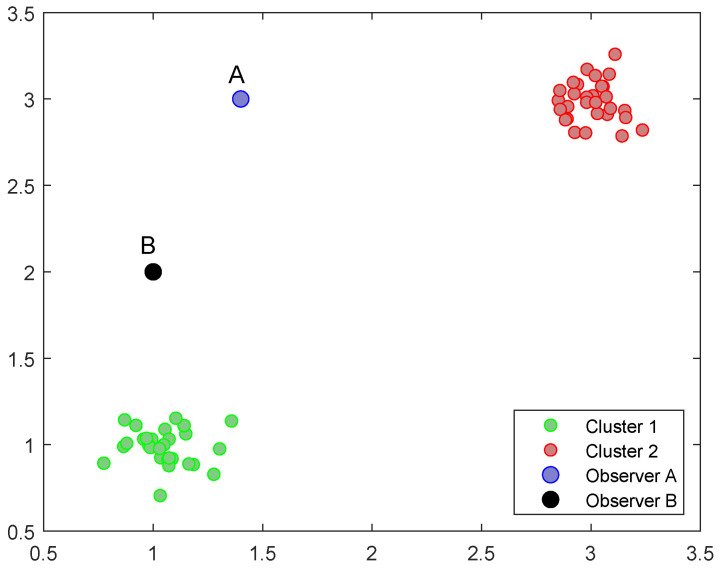
Different GMMs for different observers.

**Figure 9 sensors-24-05695-f009:**
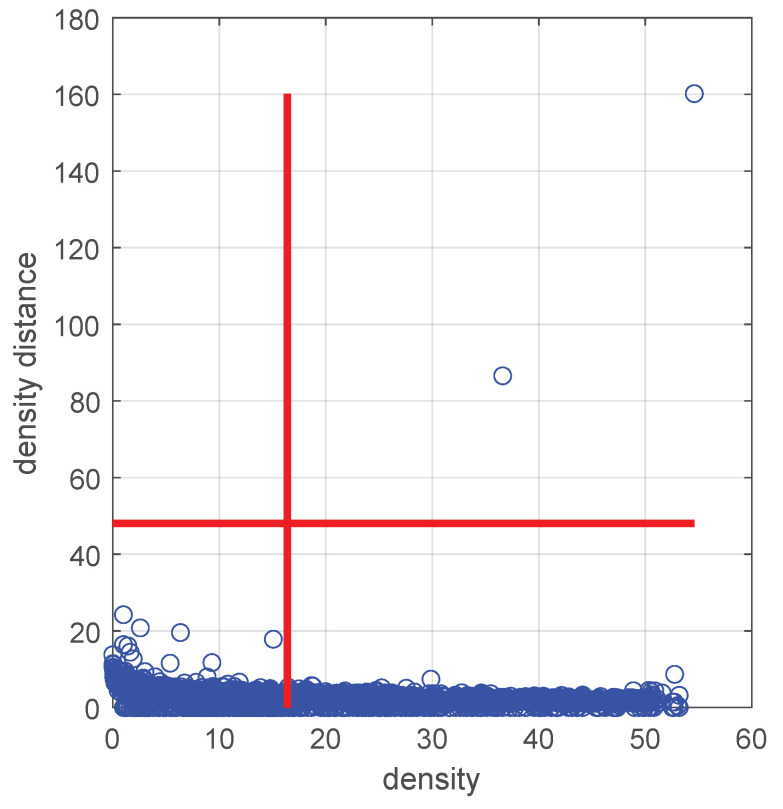
The decision graph for the third component of the optimal GMM in the KEEL Wine-White dataset.

**Figure 10 sensors-24-05695-f010:**
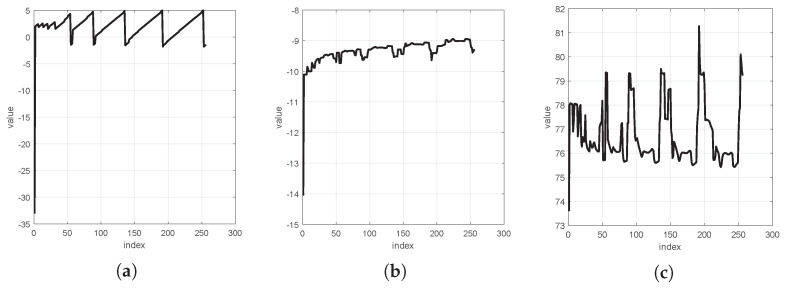
Distributions of the three dimensions in a point cloud of rock fragment surfaces: (**a**) example distribution of the X-dimension, (**b**) example distribution of the Y-dimension, and (**c**) example distribution of the Z-dimension.

**Figure 11 sensors-24-05695-f011:**
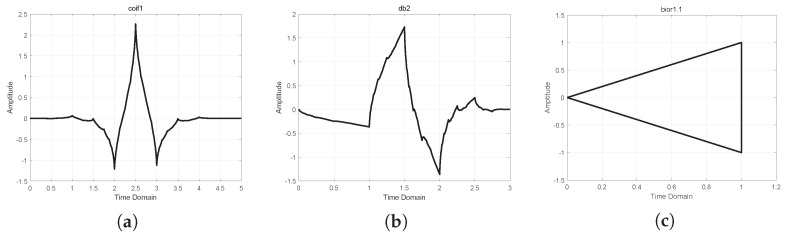
Waveforms of the three wavelets: (**a**) waveform of the coif1 wavelet, (**b**) waveform of the db2 wavelet, and (**c**) waveform of the bior1.1 wavelet.

**Figure 12 sensors-24-05695-f012:**
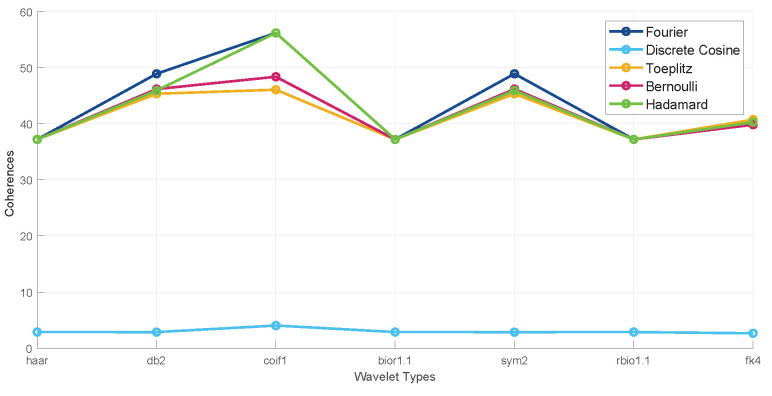
Comparison of coherence values generated by various measurement matrices and wavelet bases.

**Figure 13 sensors-24-05695-f013:**
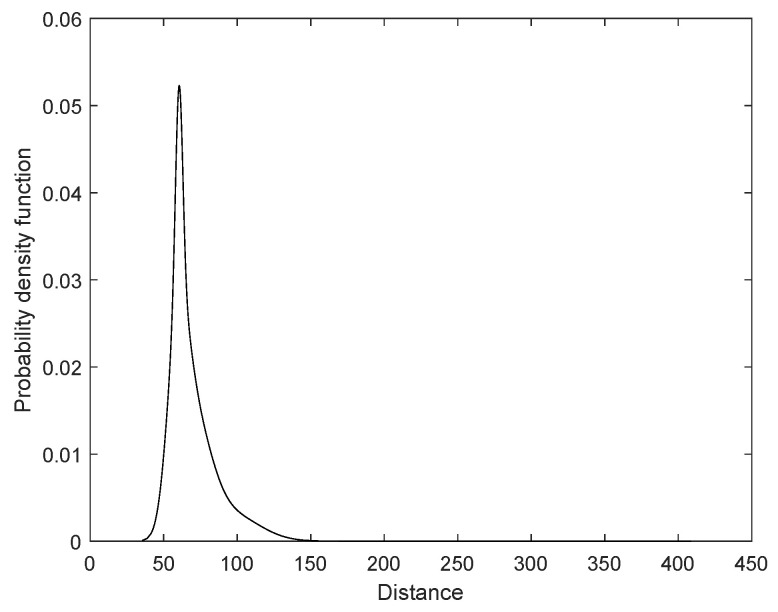
PDF of the optimal GMM.

**Figure 14 sensors-24-05695-f014:**
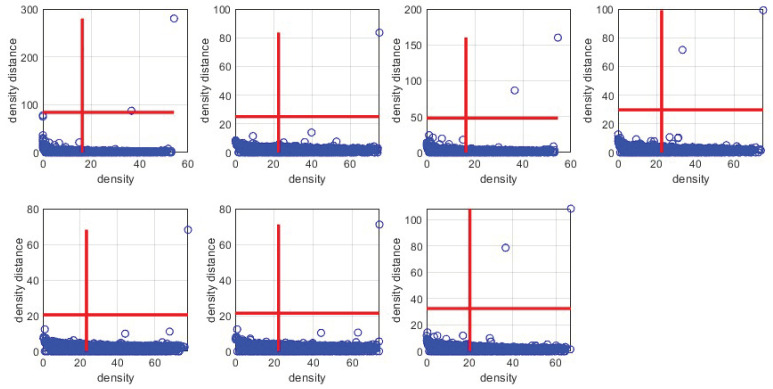
Decision graphs corresponding to each Gaussian component.

**Figure 15 sensors-24-05695-f015:**
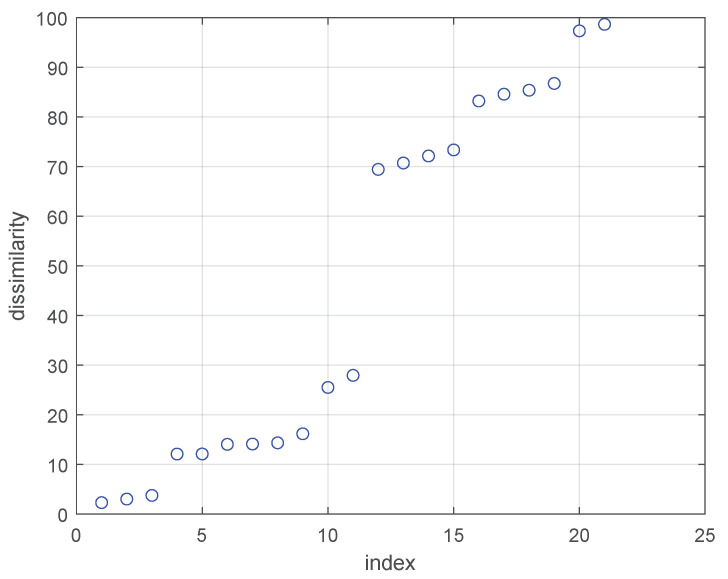
Sorted dissimilarity sequences.

**Figure 16 sensors-24-05695-f016:**
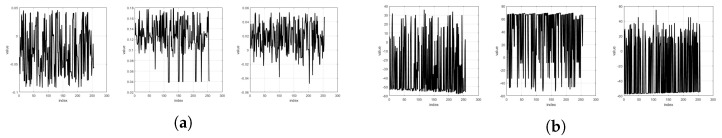
Various dimensional signal distributions of the Bunny and Armadillo point clouds: (**a**) the X-, Y-, and Z-dimensional signal distributions of the Bunny dataset and (**b**) the X-, Y-, and Z-dimensional signal distributions of the Armadillo dataset.

**Figure 17 sensors-24-05695-f017:**
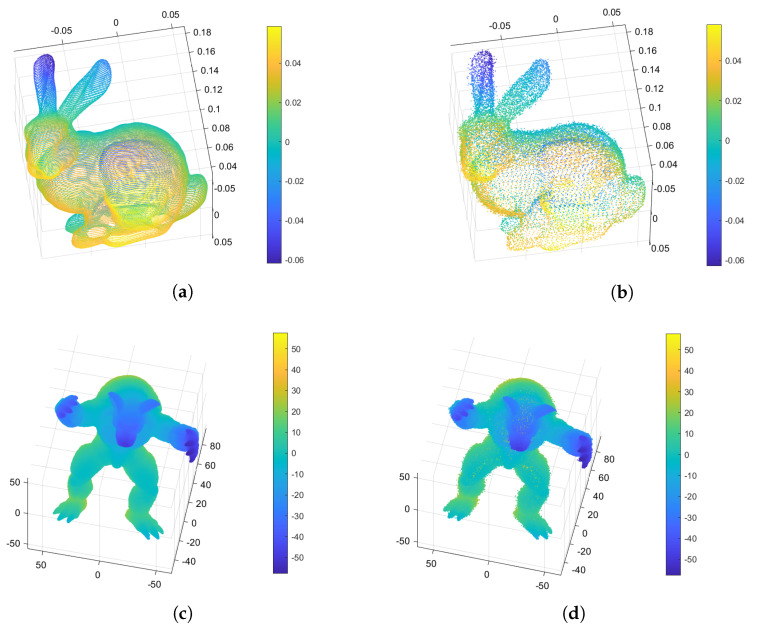
Original and DWT-based reconstructed point cloud data: (**a**) original Bunny data, (**b**) DWT-based reconstructed Bunny data, (**c**) original Armadillo data, and (**d**) DWT-based reconstructed Armadillo data.

**Figure 18 sensors-24-05695-f018:**
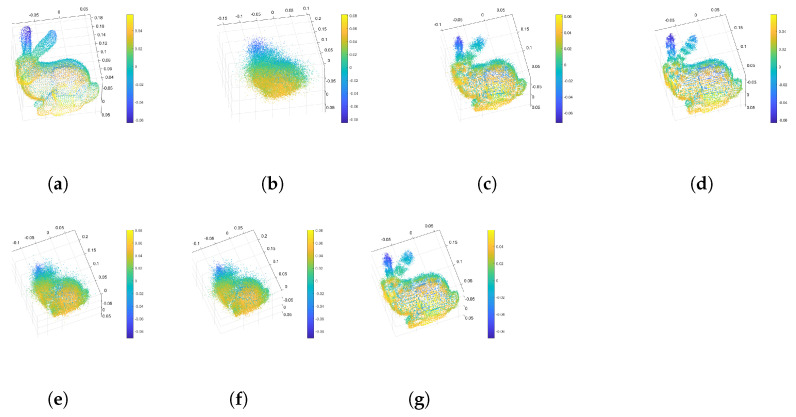
Reconstructed Bunny point clouds: (**a**) DWT-based reconstructed point cloud shape, (**b**) point cloud reconstructed using non-clustered compressive sensing, (**c**) point cloud reconstructed using our proposed approach, (**d**) point cloud reconstructed using GMM CCS, (**e**) point cloud reconstructed using Sil-based CCS, (**f**) point cloud reconstructed using CH-based CCS, (**g**) point cloud reconstructed using DB-based CCS.

**Figure 19 sensors-24-05695-f019:**
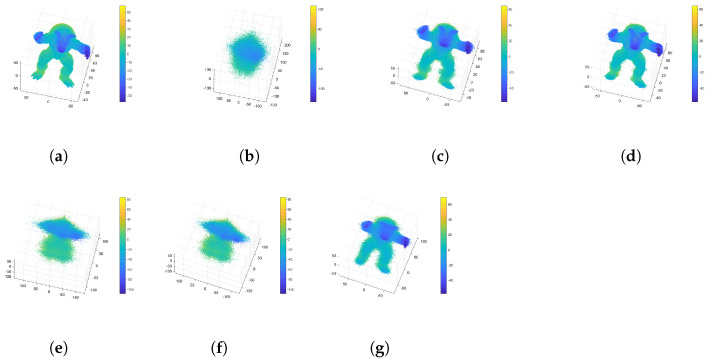
Reconstructed Armadillo point clouds: (**a**) DWT-based reconstructed point cloud shape, (**b**) point cloud reconstructed using non-clustered compressive sensing, (**c**) point cloud reconstructed using our proposed approach, (**d**) point cloud reconstructed using GMM CCS, (**e**) point cloud reconstructed using Sil-based CCS, (**f**) point cloud reconstructed using CH-based CCS, (**g**) point cloud reconstructed using DB-based CCS.

**Figure 20 sensors-24-05695-f020:**
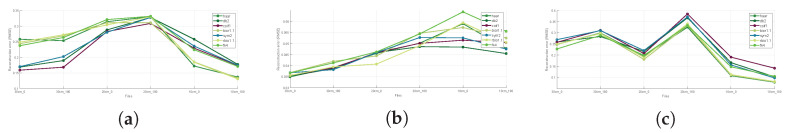
Comparison of RMSE results of different wavelets for each dimension: (**a**) X, (**b**) Y, and (**c**) Z.

**Figure 21 sensors-24-05695-f021:**
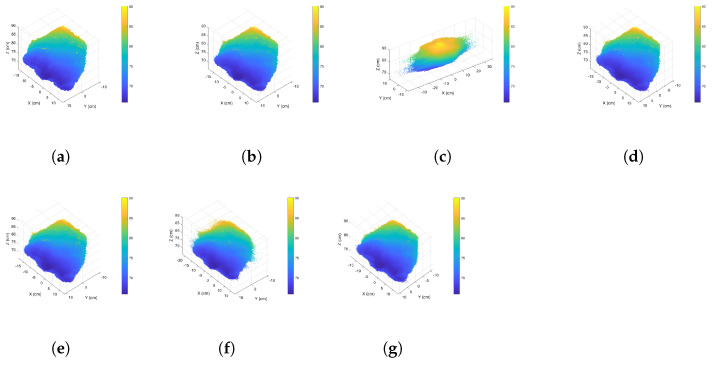
Comparative diagrams of the point cloud data shown in Figure 6a reconstructed using various compressive sensing approaches: (**a**) original data, (**b**) DWT-based data, (**c**) non-clustered compressive sensing, (**d**) our proposed CCS, (**e**) GMM-based CCS, (**f**) CH-based CCS, (**g**) DB-based CCS.

**Figure 22 sensors-24-05695-f022:**
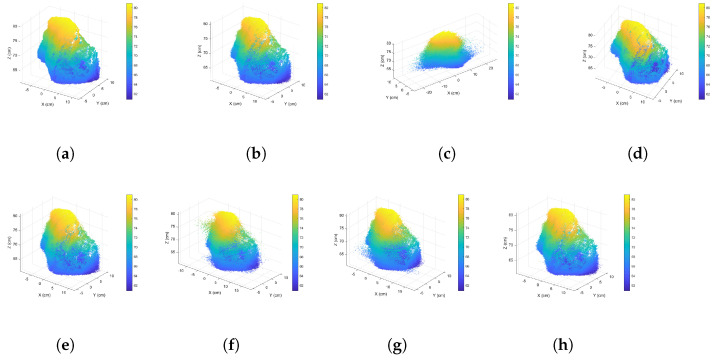
Comparative diagrams of the point cloud data shown in Figure 6b reconstructed using various compressive sensing approaches: (**a**) original data, (**b**) DWT-based data, (**c**) non-clustered compressive sensing, (**d**) our proposed CCS, (**e**) GMM-based CCS, (**f**) Sil-based CCS, (**g**) CH-based CCS, (**h**) DB-based CCS.

**Figure 23 sensors-24-05695-f023:**
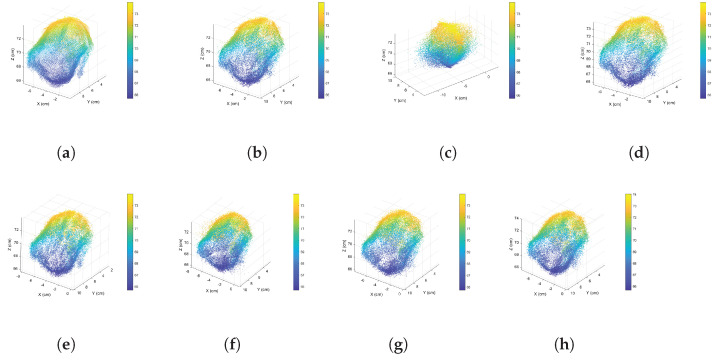
Comparative diagrams of the point cloud data shown in Figure 6c reconstructed using various compressive sensing approaches: (**a**) original data, (**b**) DWT-based data, (**c**) non-clustered compressive sensing, (**d**) our proposed CCS, (**e**) GMM-based CCS, (**f**) Sil-based CCS, (**g**) CH-based CCS, (**h**) DB-based CCS.

**Figure 24 sensors-24-05695-f024:**
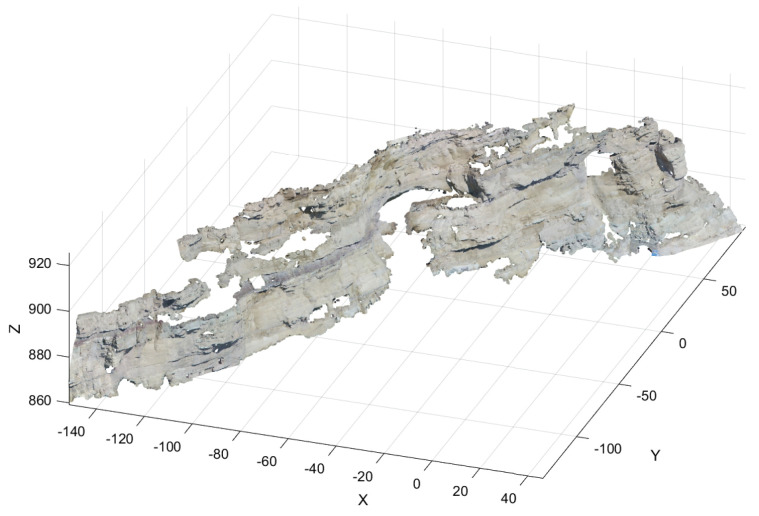
The rock outcrop point cloud data.

**Figure 25 sensors-24-05695-f025:**
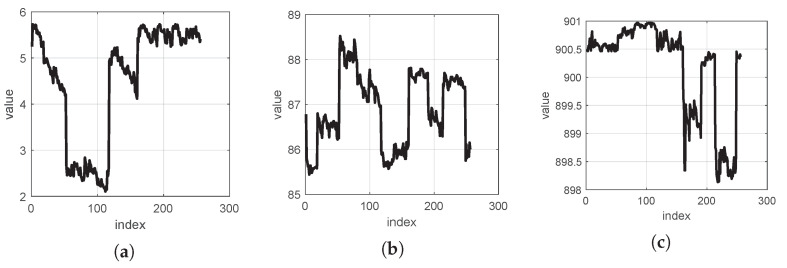
Mechanical LiDAR signal shapes of the outcrop point cloud data: (**a**) shape of the X-dimensional signal, (**b**) shape of the Y-dimensional signal, and (**c**) shape of the Z-dimensional signal.

**Table 1 sensors-24-05695-t001:** Gaussian components of the optimal GMM for the Winequality-white dataset.

Component No.	Proportion	Mean	Variance
1	0.0025	193.3033	6508.1797
2	0.1814	62.6871	50.5785
3	0.1419	96.0402	368.5748
4	0.1487	70.8337	85.0318
5	0.2290	57.8654	43.8267
6	0.1479	60.5174	6.0870
7	0.1485	76.5469	130.0642

**Table 2 sensors-24-05695-t002:** Number of clusters estimated with various clustering algorithms on four public KEEL datasets.

Dataset Name	Size	Dims	Class No.	Actual Cluster Number	GMM	Sil	CH	DB	Ours
Winequality-red	1599	11	3, 4, 5, 6, 7, 8	6	100	2	10	2	5
Winequality-white	4898	11	3, 4, 5, 6, 7, 8, 9	7	98	2	2	2	7
Texture	5500	40	2, 3, 4, 9, 10, 7, 6, 8, 12, 13, 14	11	57	99	2	99	10
Letter-recognition	20,000	16	[A-Z]	26	93	2	2	100	33

**Table 3 sensors-24-05695-t003:** Experiments on the Bunny and Armadillo point clouds.

Dataset Name	Size	Geometry Similarity	NormalSimilarity	Curvature Similarity	RMSE
Bunny	35,947	[0.4159;0.8249]	[0.0782;0.6435]	[0.0136;0.0013]	0.0011
Armadillo	172,974	[0.6948;0.8802]	[0.3038;0.8481]	[0.1768;0.2077]	0.3419

**Table 4 sensors-24-05695-t004:** PC-SSIM and RMSE of CCS (referred to as DWT-based reconstructed point cloud data).

Dataset Name	Clustering Method	Number of Clusters	Geometry Similarity	Normal Similarity	Curvature Similarity	RMSE
Bunny	∖	1	[0.3463;0.5307]	[0.0752;0.4985]	[0.0335;−0.0012]	0.0168
	Ours	16	[0.4966;0.7480]	[0.0660;0.5444]	[0.0115;$8.6887\times 10^{-4}$]	0.0035
	GMM	99	[0.5876;0.7892]	[0.0845;0.6039]	[0.0126;−0.0020]	0.0022
	Sil	2	[0.4197;0.6103]	[0.0700;0.4981]	[0.0230;0.0018]	0.01013
	CH	2	[0.4197;0.6103]	[0.0700;0.4981]	[0.0230;0.0018]	0.01013
	DB	41	[0.5156;0.7500]	[0.0753;0.5706]	[0.0110;$-2.2676\times 10^{-4}$]	0.0028
Armadillo	∖	1	[0.1983;0.3279]	[0.0790;0.4736]	[0.2510;$-1.9452\times 10^{-4}$]	21.4732
	Ours	39	[0.4719;0.6791]	[0.0876;0.5380]	[0.0882;0.0098]	2.2710
	GMM	99	[0.5359;0.7537]	[0.0873;0.5544]	[0.0620;0.0116]	1.3443
	Sil	2	[0.2410;0.3733]	[0.0801;0.4779]	[0.2465;−0.0084]	14.3123
	CH	2	[0.2410;0.3733]	[0.0801;0.4779]	[0.2465;−0.0084]	14.2876
	DB	13	[0.3722;0.5633]	[0.0842;0.5134]	[0.1324;0.0054]	1.8452

**Table 5 sensors-24-05695-t005:** Experiments on our rock fragment surface point clouds.

Dataset	Clustering Method	Number of Clusters	Geometry Similarity	Normal Similarity	Curvature Similarity	RMSE
Figure 6a	∖	1	[0.4844;0.6951]	[0.6375;0.7492]	[0.2209;$6.2052\times 10^{-4}$]	1.2975
	Ours	53	[0.7128;0.9328]	[0.6531;0.8004]	[0.3288;0.0060]	0.0955
	GMM	99	[0.7133;0.9376]	[0.6538;0.8016]	[0.3272;$2.6507\times 10^{-4}$]	0.0312
	Sil	∖	∖	∖	∖	∖
	CH	5	[0.5525;0.8395]	[0.6013;0.7121]	[0.2977;$-3.9713\times 10^{-4}$]	0.8421
	DB	57	[0.7129;0.9343]	[0.6532;0.8008]	[0.3292;0.0073]	0.0908
Figure 6b	∖	1	[0.4218;0.5916]	[0.5910;0.7188]	[0.1640;−0.0033]	1.1238
	Ours	35	[0.7293;0.9291]	[0.6360;0.8084]	[0.3290;0.0081]	0.1004
	GMM	99	[0.7334;0.9314]	[0.6369;0.8061]	[0.3247;0.0124]	0.0522
	Sil	3	[0.5169;0.8055]	[0.5736;0.6994]	[0.2525;−0.0028]	0.8783
	CH	6	[0.5247;0.8065]	[0.5746;0.6984]	[0.2489;$-6.9590\times 10^{-4}$]	0.7941
	DB	73	[0.7326;0.9304]	[0.6365;0.8083]	[0.3265;0.0157]	0.0647
Figure 6c	∖	1	[0.4412;0.5984]	[0.5854;0.7133]	[0.1650;$-2.7180\times 10^{-4}$]	0.4382
	Ours	21	[0.7067;0.9223]	[0.6254;0.8022]	[0.3248;0.0018]	0.0627
	GMM	96	[0.7178;0.9280]	[0.6255;0.8028]	[0.3309;−0.0016]	0.0413
	Sil	8	[0.5531;0.8326]	[0.5688;0.7023]	[0.2533;−0.0011]	0.2143
	CH	9	[0.5605;0.8381]	[0.5724;0.7050]	[0.2608;−0.0048]	0.1870
	DB	32	[0.7081;0.9233]	[0.6249;0.7996]	[0.3260;0.0011]	0.0564

**Table 6 sensors-24-05695-t006:** Comparison of reconstructed errors on the large-scale rock outcrop point cloud data.

Dataset	Clustering Method	Number of Clusters	Geometry Similarity	Normal Similarity	Curvature Similarity	RMSE
Figure 24 (1,844,694×3)	∖	1	[0.4896;0.6898]	[0.4634;0.7234]	[0.1252;$-9.5486\times 10^{-4}$]	0.2289
	Ours	30	[0.4887;0.6720]	[0.4602;0.7205]	[0.1130;$-2.3044\times 10^{-4}$]	0.1748

## Data Availability

The point cloud experiment data supporting this study’s findings are available in the Stanford 3D repository (https://graphics.stanford.edu/data/3Dscanrep/, accessed on 4 March 2024) and belong to the Stanford Computer Graphics Laboratory. The local clustering datasets supporting this study’s findings are available in KEEL (https://sci2s.ugr.es/keel/category.php?cat=clas, accessed on 4 March 2024; reference number 2729) and belong to the KEEL dataset repository. The authors confirm others would be able to access these data in the same manner as them and that the authors did not have any special access privileges that others would not have. The experimental datasets are available at https://doi.org/10.6084/m9.figshare.25846018. The large-scale rock outcrop point cloud is available at https://doi.org/10.6084/m9.figshare.25846210. The source codes are available at https://doi.org/10.6084/m9.figshare.25846021.

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
