# Peer review of "Compressing and Recovering Short-Range MEMS-Based LiDAR Point Clouds Based on Adaptive Clustered Compressive Sensing and Application to 3D Rock Fragment Surface Point Clouds"

_sensors, 2024, doi:10.3390/s24175695_

Round 1

Reviewer 1 Report

Comments and Suggestions for Authors

  The application of MEMS systems to 3D rock fragment surface point clouds shows considerable promise. The authors put forth an innovative MEMS-based LiDAR point cloud. This paper is entirely original and suitable for publication with only minor revisions.

1)     A comprehensive overview of the latest advancements in MEMS systems is imperative. Recently, several novel concepts have emerged in the literature, including MEMS-based ultrasensitive sensors, fractal MEMS systems, and pull-in instability, which are invaluable for this study.

     2) The use of LiDAR point clouds based on fractal theory should be highlighted, particularly the two-scale fractal dimensions, which are a valuable consideration.

      3) The English in the text requires significant improvement. Even the title is neither English nor Chinese, I suggest the following modification: Compressing and recovering short-range MEMS-based LiDAR point clouds based on adaptive clustered compressive sensing and its application to 3D rock fragment surface point clouds.

Comments on the Quality of English Language

English is poor.

Author Response

General Comment: The application of MEMS systems to 3D rock fragment surface point clouds shows considerable promise. The authors put forth an innovative MEMS-based LiDAR point cloud. This paper is entirely original and suitable for publication with only minor revisions.

Response: We appreciate your precious time reviewing our paper and providing valuable and insightful comments that made improvements in the current version possible. We have carefully considered the comments and tried our best to address every one of them. We hope the manuscript meets your high standards after careful revisions. Below, we provide the point-by-point responses. 

Comment 1: A comprehensive overview of the latest advancements in MEMS systems is imperative. Recently, several novel concepts have emerged in the literature, including MEMS-based ultrasensitive sensors, fractal MEMS systems, and pull-in instability, which are invaluable for this study.

Response 1: Thanks for your kind reminders. In our introduction, we have included an overview of the latest advancements in MEMS systems as follows: 

"Unfortunately, the high cost of high-resolution LiDAR hardware procurement impedes the popularity of this application. Recently, many advanced Micro-electro-mechanical systems (MEMS) technologies have significantly improved the manufacturing process of various sensors. The ultra-sensitive MEMS-based biosensors are applied to control pandemics at the very initial stage [3]. In the research field of the dynamic pull-in instability of a microstructure, the fractal MEMS systems are suggested to be closer to the actual state as a practical application in the air with impurities or humidity [4]. The MEMS-based LiDAR employs a micro galvanometer with only one emitter to control the light beam [5], making it more reliable due to its simple optical path structure and fewer moving parts. Therefore, Short-range MEMS (Micro-Electro-Mechanical System) based LiDAR has significantly reduced the cost of LiDAR hardware procurement from tens of thousands to just a few hundred dollars. Meanwhile, it provides dense point cloud sampling." [LN 30-42, Pg 1 of 25]

Comment 2: The use of LiDAR point clouds based on fractal theory should be highlighted, particularly the two-scale fractal dimensions, which are a valuable consideration.

Response 2: Thanks for your kind reminders. Actually, the theory of two-scale fractal dimensions gives us more inspiration. In the introduction, we highlight the importance of this theory.

"Theoretically, the two-scale fractal theory [24] represents that two scales can estimate the number of clusters: one is the scale mapping to high-dimensional data self, and the other is the observation scale mapping to an observation point with the same dimensions. Inspired by this, our innovation proposes a local clustering approach based on observation distances GMM and density peaks to obtain the number of clusters more accurately. Then, this study finds that clustered compressive sensing is utilized not only to compress and recover vibrational signals by random sampling but is appropriate for the compression and recovery of MEM-based LiDAR point clouds." [LN 93-100, Pg 2 of 25]

Comment 3: The English in the text requires significant improvement. Even the title is neither English nor Chinese, I suggest the following modification: Compressing and recovering short-range MEMS-based LiDAR point clouds based on adaptive clustered compressive sensing and its application to 3D rock fragment surface point clouds.

Response 3: Thank you very much for the kind reminders. We have updated the title to "Compressing and recovering short-range MEMS-based LiDAR point clouds based on adaptive clustered compressive sensing and its application to 3D rock fragment surface point clouds". Meanwhile, we improve the English representation of the introduction section. [LN 1, Pg 1 of 25]

We look forward to hearing from you regarding our submission. We would be glad to respond to any further questions and comments you may have. Besides, we double-checked the article throughout and corrected all typos. I appreciate your patience again.

Reviewer 2 Report

Comments and Suggestions for Authors

This study finds that clustered compressive sensing is not only utilized to compress and recover vibrational signals by random sampling but it is appropriate for the compression and recovery of MEM-based LiDAR point clouds.

I have some comments as following:

1. The title of the article is a bit long.

2. The keywords are usually 3-5 words, and the author wrote 8 words, which is a bit too many.

3. The text in Figure 1 is too small to be seen clearly.

4. The letters of images in the article, such as a, b, etc., should preferably be placed in the upper left or upper right corner of the image.

5. It is best to place Figure 16 in the center.

Author Response

General Comment: This study finds that clustered compressive sensing is not only utilized to compress and recover vibrational signals by random sampling but it is appropriate for the compression and recovery of MEM-based LiDAR point clouds.

General Response: Thank you very much for your precious time reviewing our paper and providing valuable and insightful comments that made improvements in the current version possible. We have carefully considered the comments and tried our best to address every one of them. We hope the manuscript meets your high standards after careful revisions. Below, we provide the point-by-point responses.

Comment 1: The title of the article is a bit long.

Response 1: Thanks for your kind reminders. We have carefully considered the length of the title from two aspects; one is the short-range 3D scanner topic, and the other is the applications. Therefore, the title highlights research approaches and data sources. We also note that the title is very Chinglish. The title has been updated to “Compressing and recovering short-range MEMS-based LiDAR point clouds based on adaptive clustered compressive sensing and its application to 3D rock fragment surface point clouds”. [LN 1, Pg 1 of 25]

Comment 2: The keywords are usually 3-5 words, and the author wrote 8 words, which is a bit too many.

Response 2: Thanks for your kind reminders. We have updated the keywords to “Vibrational signals; MEMS-based LiDAR; Rock fragment Surface 3D Point Cloud; Clustered compressive sensing; Local clustering”.

Comment 3: The text in Figure 1 is too small to be seen clearly.

Response 3: We apologize for the mistake. We have changed the size of the diagram as follows, and the journal should provide a high-resolution version.

Comment 4: The letters of images in the article, such as a, b, etc., should preferably be placed in the upper left or upper right corner of the image.

Response 4: We are sorry for the incorrect captions of images, and we have corrected these formats to obey the journal's requirements. These formats can be referred to: https://www.mdpi.com/1424-8220/22/24/9694 .

Comment 5: It is best to place Figure 16 in the center.

Response 5: Thanks for your kind reminders. We have updated this issue in the attached file.

We look forward to hearing from you regarding our submission. We would be glad to respond to any further questions and comments you may have. Besides, we double-checked the article throughout and corrected all typos. I appreciate your patience again.

Reviewer 3 Report

Comments and Suggestions for Authors

does the author sure about the metrics for evaluating the reconstructed point?

The author should state clearly the new contribution and insights in MEMS-based LiDAR systems?

What are the challenges face the author during the experimental setup?

How does the proposed approach compare to the other methods in terms of effectiveness?

Author Response

Comment 1: Does the author sure about the metrics for evaluating the reconstructed point?

Response 1: Thanks for your kind reminders. I am sure the metrics for evaluating the reconstructed point clouds are good. Traditionally, the RMSE is considered the metric, but the RMSE cannot reflect the morphology’s similarity of point clouds. Therefore, we use four metrics to evaluate the similarities: Geometric similarity, Normal similarity, curvature similarity, and RMSE. Other research papers have introduced these metrics, and they are referred to in our paper.

Comment 2: The author should state clearly the new contribution and insights in MEMS-based LiDAR systems.

Response 2: Thanks for your kind reminders. In our introduction, we have included an overview of the latest advancements in MEMS systems as follows: 

"Unfortunately, the high cost of high-resolution LiDAR hardware procurement impedes the popularity of this application. Recently, many advanced Micro-electro-mechanical systems (MEMS) technologies have significantly improved the manufacturing process of various sensors. The ultra-sensitive MEMS-based biosensors are applied to control pandemics at the very initial stage [3]. In the research field of the dynamic pull-in instability of a microstructure, the fractal MEMS systems are suggested to be closer to the actual state as a practical application in the air with impurities or humidity [4]. The MEMS-based LiDAR employs a micro galvanometer with only one emitter to control the light beam [5], making it more reliable due to its simple optical path structure and fewer moving parts. Therefore, Short-range MEMS (Micro-Electro-Mechanical System) based LiDAR has significantly reduced the cost of LiDAR hardware procurement from tens of thousands to just a few hundred dollars. Meanwhile, it provides dense point cloud sampling." [LN 30-42, Pg 1 of 25]

Comment 3: What are the challenges face the author during the experimental setup?

Response 3: Thanks for your kind reminders. During the experimental setup, our parameters are reliable based on the probabilities of the data. The density peaks algorithm is introduced in our paper, and it provides the typical parameters. Meanwhile, the algorithm suggests that the reduced distance can be estimated by the times of the distances’ variance. We have referred to Rodriguez’s paper (Clustering by fast search and find of density peaks), so the challenges of the experimental setup are mentioned in our paper.

Comment 4: How does the proposed approach compare to the other methods in terms of effectiveness?

Response 4: Thanks for your kind reminders. It is an exact issue in our paper. We demonstrate the better performance of our innovative method compared to indices-based approaches such as Silhouette (Sil), CalinskiHarabasz (CH), and DaviesBouldin (DB) and append these experiments into our paper as follows:

We separately estimate the number of clusters using our proposed method and the traditional GMM. Table. 4 shows the PC-SSIM and RMSE in the non-clustered, traditional GMM clustered, and our proposed CCS method referred to the DWT-based reconstructed point cloud data. Meanwhile, we also compare indices-based CCS methods, which utilize Silhouette, CalinskiHarabasz, and DaviesBouldin indices as clustering analysis metrics. The sampling ratio is set to 0.4. "\" denotes the non-clustered compressive sensing method. [LN 452-457, Pg 15 of 25]

More details are represented in the attached.

We look forward to hearing from you regarding our submission. We would be glad to respond to any further questions and comments you may have. Besides, we double-checked the article throughout and corrected all typos. I appreciate your patience again.

Reviewer 4 Report

Comments and Suggestions for Authors

Vibrational noise prevents compressive sensing in many applications, and this method seems to be suitable also to such SLAM problems, where key points tend to have natural errors (e.g. SLAM in the forest environment). That is why this method may have more application cases, where the execution time is not crucial. 

The structure of the paper is clear and citations cover the field well. The tests used are sound although some measures used might require an explanation (although those can be found in a reference). The results can be reproduced, since both the data and algorithms are available. Visualizations civer well the aspects of data and the compressive sensing problem. Conclusion sums up the work well. 

General comment: the quality of the article seems to be right after earlier iterations? A positive point is that all the data and code is available! 

Detailed comments: 

RIP == restricted isometric property, not restrict isometric property (this may be corrected by editors)

About the conclusion: Defining the cluster number is not a big problem in practise. One can tune it to get a satisfactory result when a relatively monotonic samples can be expected.  

Author Response

General Comment: The quality of the article seems to be right after earlier iterations? A positive point is that all the data and code is available! 

General Response: Thanks for your positive comment. Some professors are concerned about our findings on vibrational signal handling. Still, others are more interested in estimating the number of clusters closer to the actual because of the large size of dense point clouds. The first submission is built on two rounds of modification. I guess that more deep-learning applications of dense point clouds should be required for the estimation so that some algorithms evaluate the lower bound of the fusion model under the transformer neural network. Therefore, we study both aspects for some applications, such as compression, reconstruction, and up-sampling.

Comment 1: RIP == restricted isometric property, not restrict isometric property (this may be corrected by editors).

Response 1: Thank you very much. We apologize for this mistake. We have modified “restrict” to “restricted.” [LN 73, Pg 2 of 25]

Comment 2: About the conclusion: Defining the cluster number is not a big problem in practise. One can tune it to get a satisfactory result when a relatively monotonic samples can be expected.

Response 2: Agree. Conveniently, the number of clusters can be obtained with a simple tune if the size of point clouds is small. However, dense point clouds are very large, and the artificial tune is low effective. Therefore, we provide an effective approach to obtain the number of clusters closer to the actual. This approach can also be applied to selecting parameters in the fusion model based on the Transformer Neural Network model.

. We look forward to hearing from you regarding our submission. We would be glad to respond to any further questions and comments you may have. Besides, we double-checked the article throughout and corrected all typos. I appreciate your patience again.
